# Cross-Modal Semantic Decoupling and Transfer for Text-to-Visible-Infrared Person Re-Identification

Ziang Zhang [* 1]   Bin Yang [* 1]   Mang Ye [1]

https://github.com/witpol96/CSDT

## Abstract

Text-to-Image Person Re-Identification (TI-ReID) retrieves visible pedestrian images using text queries. Yet in low-light or nighttime settings, visible images lack sufficient identity details, while infrared images effectively capture pedestrian contours and textures. To enable all-day surveillance, we propose a dual cross-modal retrieval task called Text-to-Visible-Infrared Re-Identification (TVI-ReID) and construct corresponding tri-modal datasets. Compared to TI-ReID, TVI-ReID faces two key challenges: (1) *complex hybrid discrepancies* in dual cross-modal retrieval from three modalities, and (2) *semantic inconsistency* between pretraining and downstream tasks. To address these issues, we propose a **C**ross-Modal **S**emantic **D**ecoupling and **T**ransfer (CSDT) framework. CSDT constructs color-related and color-irrelevant feature subspaces via Semantic Decoupling Learning (SDL) to align shared semantics across text and dual image modalities, reducing hybrid discrepancies. Moreover, Semantic Distribution Transfer (SDT) adapts pretrained text-visible alignment to text-infrared matching. Extensive experiments on tri-modal datasets show our approach outperforms existing state-of-the-art TI-ReID methods.

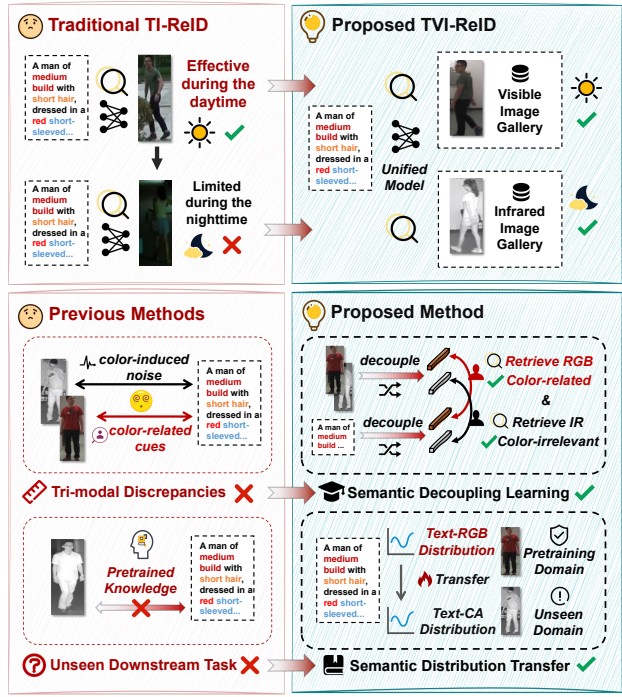

*Figure 1.* Illustration of the motivation. Previous TI-ReID methods mainly focus on retrieving visible images, which is limited in nighttime scenarios. To address this issue, we propose Text-to-Visible-Infrared Re-identification (TVI-ReID), which jointly retrieves visible and infrared images to enable all-day surveillance.

## 1. Introduction

Text-to-Image Person Re-Identification (TI-ReID) is a cross-modal retrieval task that aims to match pedestrian images with their corresponding textual descriptions (Li et al., 2017). As shown in the top-left of Fig. 1, current text-to-image person re-identification frameworks mainly focus on retrieving

visible pedestrian images. However, under nighttime or low-light conditions, visible images fail to capture discriminative identity information of pedestrians, leading to a sharp degradation in cross-modal retrieval performance. As a result, these methods struggle to meet the demands of retrieving nighttime active targets with eyewitness descriptions.

Since infrared cameras can capture pedestrian contours and texture information under low-light conditions (Wu et al., 2017; Nguyen et al., 2017; Zhang & Wang, 2023), extending TI-ReID to text-to-visible-infrared image retrieval is therefore crucial for 24-hour surveillance. To this end, we propose the task of **Text-to-Visible-Infrared Person Re-Identification (TVI-ReID)**, as shown in the top-right of

---

[*]Equal contribution   [1]School of Computer Science, Wuhan University, Wuhan, China. Correspondence to: Mang Ye <ye-mang@whu.edu.cn>.

*Proceedings of the 43rd International Conference on Machine Learning*, Seoul, South Korea. PMLR 306, 2026. Copyright 2026 by the author(s).

Fig. 1, which aims to retrieve person images from both visible and infrared image galleries with language descriptions. Based on existing VI-ReID and omni multi-modal ReID datasets (Wu et al., 2017; Zhang & Wang, 2023; Zuo et al., 2026), we manually annotated and constructed a new tri-modal dataset **SYSU-TVI** and organized another dataset **LLCM-TVI** to support the TVI-ReID task.

Existing text-to-image person ReID frameworks mainly rely on contrastive learning and metric learning to align shared identity-related attributes across heterogeneous modalities (Bai et al., 2023; Cao et al., 2024; Jiang & Ye, 2023; Qin et al., 2024), such as color, in order to construct a cross-modal feature space with strong discriminative capability. However, the discrepancies across multiple modalities make it challenging for TI-ReID model to effectively capture identity-related information in both text-to-visible and text-to-infrared retrieval tasks simultaneously. For instance, clothing color is crucial for visible-image retrieval, but it is identity-irrelevant and should be eliminated as interference in text-to-infrared retrieval. This raises a pivotal problem: ***I) how can we flexibly extract shared identity-related information for both text-to-visible and text-to-infrared retrieval tasks, while suppressing interference from identity-irrelevant information?***

With the rapid development of vision-language pretraining, fine-tuning general-purpose pretrained models (Radford et al., 2021) or TI-ReID pretrained models (Tan et al., 2024; Jiang et al., 2025) has become the main approach in TI-ReID. However, during pretraining, the heavy involvement of visible images establishes strong correlations between entity semantics and specific color types (e.g., "apple–red," "jeans–blue," "leather shoes–brown"). Infrared or single-channel images exhibit significant differences from real-world natural entities in terms of color characteristics, which severely limits the transferability of conventional fine-tuning paradigms to text-to-infrared downstream task. This further raises the second key question: ***II) how can the retrieval capabilities of pretrained models be effectively transferred to unseen text-to-infrared retrieval task?***

To address the above issues, we propose the **Cross-Modal Semantic Decoupling and Transfer (CSDT)** framework to enable robust joint cross-modal retrieval. For problem ***I)***, we propose the **Semantic Decoupled Learning (SDL)** paradigm to decouple the shared tri-modal feature space into color-related and color-irrelevant feature subspaces, thereby simultaneously reducing the cross-modal discrepancies between text and dual-spectral images. Compared to traditional multi-modal ReID methods (Li et al., 2024; Zuo et al., 2026), which focus on constructing a unified cross-modal feature space, SDL performs cross-modal metric learning separately in two orthogonal subspaces, effectively preventing the introduction of identity-irrelevant information when

retrieving multiple heterogeneous modalities. For problem ***II)***, we propose the **Semantic Distribution Transfer (SDT)** strategy, which transfers knowledge by minimizing the KL divergence between the text-image similarity distributions of color-related and color-irrelevant domains. Compared to the traditional fine-tuning strategy in TI-ReID (Jiang & Ye, 2023; Zuo et al., 2024), SDT uses the color-related text-visible similarity distribution as a supervisory signal, which reflects the strength of pretrained cross-modal semantic correlations, to guide contrastive learning of text and single-channel images in color-irrelevant feature subspace, thus significantly improving the model's transferability in text-to-infrared retrieval task. **CSDT** integrates **SDL** and **SDT** to perform cross-modal semantic decoupling and transfer, effectively mitigating the heterogeneous modality gap and facilitating efficient cross-modal retrieval.

Our main contributions can be summarized as follows:

- **Task Contribution.** We propose a new and important task called Text-to-Visible-Infrared Re-Identification (TVI-ReID) to address the limitation of TI-ReID in nighttime performance. Meanwhile, based on existing datasets, we construct tri-modal datasets SYSU-TVI and LLCM-TVI to support the research of TVI-ReID.

- **Methodological Contribution.** We propose the Cross-Modal Semantic Decoupling and Transfer (CSDT) framework, which consists of two modules: Semantic Decoupling Learning (SDL) module aligns color-related and color-irrelevant identity features across different modalities, while Semantic Distribution Transfer (SDT) module transfers pretrained knowledge to the unseen downstream text-to-infrared retrieval task.

- **Empirical Contribution.** Extensive experiments on SYSU-TVI and LLCM-TVI demonstrate the effectiveness of our model in performing retrieval across different modalities, surpassing state-of-the-art TI-ReID models on TVI-ReID retrieval task.

## 2. Related Works

### 2.1. Text-to-image Re-Identification

Text-to-Image Person Re-Identification (TI-ReID) (Li et al., 2017) is a challenging cross-modal retrieval task. Given a textual pedestrian description, the model is required to retrieve visible (RGB) pedestrian images with the same identity. The early approaches can be mainly categorized into global matching (Zhang & Lu, 2018; Li et al., 2017) and local matching (Wang et al., 2020; Wu et al., 2021b; Chen et al., 2022b; Yan et al., 2023) methods. Recently, many studies have focused on fine-tuning vision-language pretrained models (e.g., CLIP (Radford et al., 2021)) to fully leverage their rich cross-modal pretrained knowledge (Jiang

& Ye, 2023; Qin et al., 2024; Zhao et al., 2024). At the same time, several works have begun exploring pretrained models specifically for TI-ReID to build robust general TI-ReID models with strong robustness across domains (Zuo et al., 2024; Tan et al., 2024; Jiang et al., 2025; Wang et al., 2025c;a). However, although these works have achieved impressive performance on existing TI-ReID datasets such as CUHK-PEDES (Li et al., 2017), ICFG-PEDES (Ding et al., 2021), and RSTPReid (Zhu et al., 2021), they still struggle to effectively retrieve pedestrian images under nighttime conditions. Deng et al (Deng et al., 2025) proposed text-RGBT person retrieval to address this problem. However, it is limited to retrieving paired fused visible-thermal images, and the captured pedestrian information remains insufficient. In this paper, we propose the TVI-ReID task, which introduces infrared images into the TI-ReID retrieval task, thus extending TI-ReID to all-day (24-hour) surveillance.

## 2.2. Visible-Infrared Re-Identification

The goal of Visible-Infrared Person Re-Identification (VI-ReID) is to retrieve pedestrian images across visible and infrared modalities under different settings (Wu et al., 2017; Nguyen et al., 2017; Dai et al., 2018; Wu et al., 2021a; Zhang & Wang, 2023; Yang et al., 2022; 2023; Yao et al., 2025; Wang et al., 2024; Zhou et al., 2020). Compared to traditional ReID methods (Ye et al., 2021b; He et al., 2021; Ye et al., 2025; Li et al., 2022; 2020; Dong et al., 2024; Lu et al., 2025), VI-ReID methods mainly focus on mining modality-shared identity information from features extracted by CNNs or Transformers (Wang et al., 2019; Zhang et al., 2022a;b; Chen et al., 2022a; Wang et al., 2025b). Recently, text and vision-language pretrained models (Radford et al., 2021) have been increasingly used to assist in aligning heterogeneous image modalities in VI-ReID. Some works attempt to leverage text and vision-language pretrained models to guide visible and infrared images in focusing on aligning high-level semantic information, thereby mitigating the cross-modal gap caused by differences in low-level visual features (Yu et al., 2025). Meanwhile, other approaches use text as auxiliary information in retrieval process, compensating for the lack of critical color-related information in the infrared modality to further bridge the gap with visible images (Du et al., 2024; Hu et al., 2024). In contrast to these previous works, the proposed TVI-ReID treats both visible and infrared images as retrieval targets for text descriptions, which introduces additional challenges in cross-modal alignment due to modality gaps across multiple semantic levels.

## 3. Proposed Method

**Formulation.** Compared to previous Text-to-Image Person Re-identification tasks, the TVI-ReID task introduces more complex modality relationships. Let the sample set for

TVI-ReID be denoted as $\{\mathcal{D}^{v,rgb}, \mathcal{D}^{v,ir}, \mathcal{D}^{t,rgb}, \mathcal{D}^{t,ir}, \mathcal{Y}\}$, where $\mathcal{D}^{v,rgb}$ represents the visible image data and $\mathcal{D}^{t,rgb}$ represents its corresponding textual data. Similarly, $\mathcal{D}^{v,ir}$ and $\mathcal{D}^{t,ir}$ represent the infrared image data and its corresponding textual data, respectively, and $\mathcal{Y}$ denotes the label set. It should be noted that the data for $\mathcal{D}^{t,ir}$ are also annotated based on **visible images** of pedestrians with the same identity, to align with the descriptions of eyewitnesses in real-world scenarios. During the inference process, the model uses 2 query sets and 2 gallery sets for retrieval. $\mathcal{Q}^{rgb}$ and $\mathcal{Q}^{ir}$ correspond to $\mathcal{D}^{t,rgb}$ and $\mathcal{D}^{t,ir}$, respectively, while $\mathcal{G}^{rgb}$ and $\mathcal{G}^{ir}$ correspond to $\mathcal{D}^{v,rgb}$ and $\mathcal{D}^{v,ir}$.

**Overview.** As illustrated in Fig. 2, CSDT adopts a multi-branch architecture to separately extract color-related and color-irrelevant features from both text and images. For visible images, CSDT encodes the RGB three-channel images using the color-related branch, while simultaneously employing the color-irrelevant branch to encode their corresponding single-channel grayscale images (Ye et al., 2021a). For text, CSDT also extracts color-related and color-irrelevant features using different branches. For infrared images, CSDT directly extracts color-irrelevant features using the color-irrelevant branch. Subsequently, CSDT incorporates two core modules, **Semantic Decoupling Learning (SDL)** and **Semantic Distribution Transfer (SDT)**, to achieve color semantic decoupling and transfer in multi-modal joint training. **SDL** performs metric learning on text–image features extracted by different branches to construct color-related and color-irrelevant feature subspaces and enforces an orthogonal loss to minimize the mutual information between the two subspaces, thus reducing identity-irrelevant interference from color information. Meanwhile, **SDT** employs the text-RGB similarity distribution as soft labels to guide the alignment between text and single-channel image representations. effectively transferring the pretrained knowledge of cross-modal alignment to the subspace of color-relevant features and improving the performance in downstream text-to-infrared retrieval task.

## 3.1. Semantic Decoupling Learning

The optimization goal of traditional TI-ReID is to ensure that the anchor feature is closer to the positive features than to the negative features in cross-modal feature space. By reducing the distance between cross-modal positive sample pairs, the discrepancy across different modalities can be eliminated, thereby facilitating the learning of identity-discriminative features. The optimization goal can be expressed as follows:

$$D(f^{rgb,a}, f^{t,p}) < D(f^{rgb,a}, f^{t,n}), \quad (1)$$

$$D(f^{t,a}, f^{rgb,p}) < D(f^{t,a}, f^{rgb,n}), \quad (2)$$

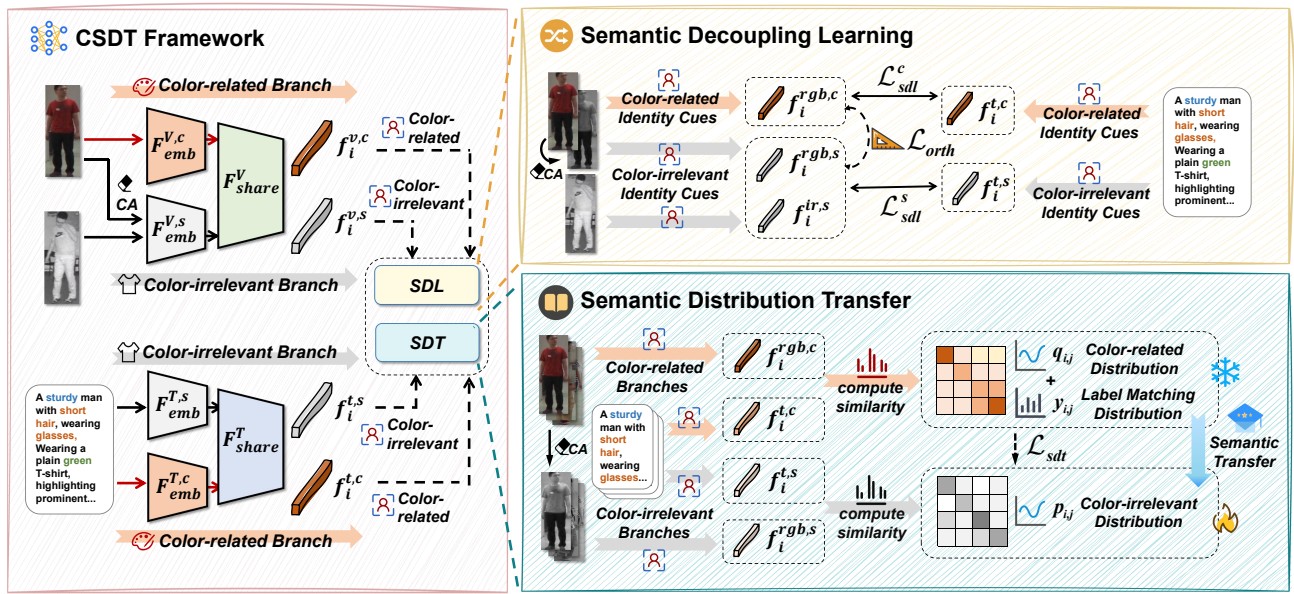

*Figure 2.* Illustration of our **CSDT**. CSDT adopts a multi-branch architecture to decouple color-related and color-irrelevant features. Meanwhile, the **Semantic Decoupling Learning (SDL)** module helps the model align text-image features in two orthogonal subspaces, while the **Semantic Distribution Transfer (SDT)** module facilitates the transfer of pretrained knowledge to unseen downstream task.

where $D$ denotes the distance function (e.g., cosine similarity distance), $f^{t,a}$, $f^{t,p}$ and $f^{t,n}$ denote the anchor, positive and negative features of the text modality, $f^{rgb,a}$, $f^{rgb,p}$ and $f^{rgb,n}$ denote the anchor, positive and negative features of the visible image modality. However, for TVI-ReID, we also need to optimize the model to satisfy the following objectives:

$$D(f^{ir,a}, f^{t,p}) < D(f^{ir,a}, f^{t,n}), \tag{3}$$

$$D(f^{t,a}, f^{ir,p}) < D(f^{t,a}, f^{ir,n}), \tag{4}$$

where $f^{ir,a}$, $f^{ir,p}$ and $f^{ir,n}$ denote the anchor, positive and negative features of infrared image modality.

Previous TI-ReID methods typically adopt metric learning to learn a shared image-text identity-discriminative feature space $\mathcal{F}$, under which the distance function $D$ is expected to reliably measure cross-modal identity distance, satisfying Eq. 1 and Eq. 2. However, learning a tri-modal shared feature space $\mathcal{F}$ for TVI-ReID which simultaneously satisfies Eq. 1– 4 remains highly challenging. This difficulty arises from the tri-modal discrepancies: color-related information (e.g., clothing color) in visible images and textual descriptions is highly correlated with pedestrian identity and crucial for measuring cross-modal feature distances, whereas infrared images primarily rely on color-irrelevant information (e.g., clothing style, body shape, and accessories), requiring the suppression of color-related interference. To address this issue, we adopt **Semantic Decoupling Learning** to decouple the feature space into the **color-related subspace** $\mathcal{F}^c$ and the **color-irrelevant subspace** $\mathcal{F}^s$, as shown in Fig. 2.

In subspace $\mathcal{F}^c$, we align the color-related features of text and images to satisfy Eq. 1 and Eq. 2, while in subspace $\mathcal{F}^s$, we align the color-irrelevant features of text and images to satisfy Eq. 3 and Eq. 4.

Many studies have demonstrated that even subtle variations in shallow token embeddings can have a significant impact on multi-modal semantic representation learning (Zhou et al., 2022; Yang et al., 2024). Motivated by this observation, we copied both the image and text shallow tokenizers and extent CLIP into the **multi-branch architecture**. Different branches are respectively designed to extract color-related and color-irrelevant features from different modalities, as illustrated in Fig. 2, where $F_{emb}^{V,c}$ and $F_{emb}^{V,s}$ denote the color-related and color-irrelevant tokenizers for visual encoder, while $F_{emb}^{T,c}$ and $F_{emb}^{T,s}$ denote the color-related and color-irrelevant tokenizers for text encoder. Meanwhile, color-related and color-irrelevant branches share the same deep Transformer blocks (i.e., $F_{share}^V$ and $F_{share}^T$) to model the corresponding high-level semantic representations (Vaswani et al., 2017; Dosovitskiy et al., 2020).

Specifically, within each training batch, we extract both color-related and color-irrelevant features from visible images. The color-related branch directly encodes the visible images to capture color-related identity cues. Meanwhile, for the same visible images, we remove color information using Channel-Augmentation (Ye et al., 2021a) technique and encode the resulting colorless single-channel images with the color-irrelevant branch to capture color-irrelevant identity cues in the same visible images. The above process

can be formulated as follows:

$$f_i^{rgb,c} = F_{share}^V(F_{emb}^{V,c}(V_i^{rgb})), \tag{5}$$

$$f_i^{rgb,s} = F_{share}^V(F_{emb}^{V,s}(V_i^k)), \tag{6}$$

$$V_i^k \sim \{V_i^r, V_i^g, V_i^b\} \tag{7}$$

where $V_i^{rgb}$, $V_i^r$, $V_i^g$ and $V_i^b$ denote the visible RGB image, the R-channel image, the G-channel image, and the B-channel image, respectively, $f^{rgb,c}$ and $f^{rgb,s}$ represent the color-related and color-irrelevant features of visible images, $\sim$ denotes random selection. During the Channel-Augmentation operation, we randomly select one single-channel image from $V_i^r$, $V_i^g$ and $V_i^b$ as the input of color-irrelevant branch, in order to eliminate the influence of color information in contrastive learning. For text, we directly extract features using both the color-related and color-irrelevant branches, which can be formulated as follows:

$$f_i^{t,c} = F_{share}^T(F_{emb}^{T,c}(T_i)), \tag{8}$$

$$f_i^{t,s} = F_{share}^T(F_{emb}^{T,s}(T_i)), \tag{9}$$

where $T_i$ denotes the text description, $f_i^{t,c}$ and $f_i^{t,s}$ represent the color-related and color-irrelevant features of text. Subsequently, we employ the triplet alignment loss (Qin et al., 2024) to construct feature subspaces $\mathcal{F}^c$ and $\mathcal{F}^s$ with strong identity discriminability, which can be formulated as follows:

$$\mathcal{L}_{sdl}^c = [m - \sum_{j=1}^{N^+} S(f_i^{rgb,c}, f_j^{t,c+}) + \sum_{j=1}^{N^-} S^*(f_i^{rgb,c}, f_j^{t,c-})]_+$$

$$+ [m - \sum_{j=1}^{N^+} S(f_i^{t,c}, f_j^{rgb,c+}) + \sum_{j=1}^{N^-} S^*(f_i^{t,c}, f_j^{rgb,c-})]_+, \tag{10}$$

$$\mathcal{L}_{sdl}^s = [m - \sum_{j=1}^{N^+} S(f_i^{rgb,s}, f_j^{t,s+}) + \sum_{j=1}^{N^-} S^*(f_i^{rgb,s}, f_j^{t,s-})]_+$$

$$+ [m - \sum_{j=1}^{N^+} S(f_i^{t,s}, f_j^{rgb,s+}) + \sum_{j=1}^{N^-} S^*(f_i^{t,s}, f_j^{rgb,s-})]_+, \tag{11}$$

where $m$ is a positive margin coefficient, $S$ denotes cosine similarity, $S^*$ denotes the weighted negative similarity (Qin et al., 2024), $[x]_+ = max(x, 0)$, $N^+$ and $N^-$ denote the number of positive and negative samples, $f_j^{rgb,c+}$, $f_j^{t,c+}$, $f_j^{rgb,s+}$ and $f_j^{t,s+}$ denote the positive samples of different modalities, $f_j^{rgb,c-}$, $f_j^{t,c-}$, $f_j^{rgb,s-}$ and $f_j^{t,s-}$ denote the negative samples of different modalities. To enhance the independence between the two feature subspaces (Zhang et al., 2024), we employ an orthogonal loss to constrain color-related feature and color-irrelevant feature extracted

from the same image to remain orthogonal, which can be expressed as follows:

$$\mathcal{L}_{orth} = \frac{1}{N} \sum_{i=1}^N [\frac{\langle f_i^{rgb,c}, f_i^{rgb,s} \rangle}{\|f_i^{rgb,c}\|_2 \cdot \|f_i^{rgb,s}\|_2}]_+, \tag{12}$$

where $\langle \cdot, \cdot \rangle$ denotes the dot product of two embeddings, $\|\cdot\|_2$ denotes the L2 distance.

Based on the prior knowledge that color information is identity-irrelevant for text-to-infrared retrieval, we extract color-irrelevant features from infrared images using only the color-irrelevant branch, which can be formulated as follows:

$$f_i^{ir,s} = F_{share}^V(F_{emb}^{V,s}(V_i^{ir})), \tag{13}$$

where $V_i^{ir}$ denotes the infrared image, $f_i^{ir,s}$ represents the color-irrelevant feature of infrared image. To enable text and infrared images to satisfy the constraint relationships Eq.3 and Eq.4 in the color-irrelevant feature space $\mathcal{F}^s$, we jointly optimize them with the color-irrelevant features of visible images using the triplet alignment loss. Accordingly, we modify the loss function $\mathcal{L}_{sdl}^s$ in Eq.11 as follows:

$$\mathcal{L}_{sdl}^{s*} = [m - \sum_{j=1}^{N^+} S(f_i^{ir,s}, f_j^{t,s+}) + \sum_{j=1}^{N^-} S^*(f_i^{ir,s}, f_j^{t,s-})]_+$$

$$+ [m - \sum_{j=1}^{N^+} S(f_i^{t,s}, f_j^{ir,s+}) + \sum_{j=1}^{N^-} S^*(f_i^{t,s}, f_j^{ir,s-})]_+$$

$$+ [m - \sum_{j=1}^{N^+} S(f_i^{t,s}, f_j^{rgb,s+}) + \sum_{j=1}^{N^-} S^*(f_i^{t,s}, f_j^{rgb,s-})]_+. \tag{14}$$

Finally, SDL module jointly optimizes $\mathcal{L}_{sdl}^c$ and $\mathcal{L}_{sdl}^{s*}$ to simultaneously align text-image features in the two feature subspaces, which can be expressed as follows:

$$\mathcal{L}_{sdl} = \mathcal{L}_{sdl}^c + \mathcal{L}_{sdl}^{s*} + \mathcal{L}_{orth}. \tag{15}$$

### 3.2. Semantic Distribution Transfer

The SDL module effectively mitigates performance bias caused by tri-modal discrepancies in different retrieval tasks. However, due to the model's emphasis on color information during pretraining (Radford et al., 2021), existing cross-modal alignment knowledge is not fully leveraged when constructing $\mathcal{F}^s$. To address this problem, we propose the **Semantic Distribution Transfer**, which aims to transfer the text-image alignment relationship from $\mathcal{F}^c$ to $\mathcal{F}^s$, thereby enhancing the cross-modal discriminability of $\mathcal{F}^s$ and improving the performance of text-to-infrared retrieval task.

Specifically, for each visible image in a training batch, we compute its color-related similarity in $\mathcal{F}^c$ and color-irrelevant similarity in $\mathcal{F}^s$ with all text samples in the batch.

For each text-image pair, the similarities can be computed as follows:

$$p_{i,j} = \frac{exp(S(f_i^{rgb,s}, f_j^{t,s})/\tau)}{\sum_k exp(S(f_i^{rgb,s}, f_k^{t,s})/\tau)}, \quad (16)$$

$$q_{i,j} = \frac{exp(S(f_i^{rgb,c}, f_j^{t,c})/\tau)}{\sum_k exp(S(f_i^{rgb,c}, f_k^{t,c})/\tau)}, \quad (17)$$

where $p_{i,j}$ and $q_{i,j}$ denotes the similarities in $\mathcal{F}^s$ and $\mathcal{F}^c$, and $\tau$ is a tunable temperature parameter. Notably, when computing color-irrelevant similarity $p_{i,j}$, we substitute $f_i^{ir,s}$ with single-channel $f_i^{rgb,s}$ to mitigate noise induced by non-paired visible-infrared image viewpoints. Then we optimize the distance between $q_{i,j}$ and $p_{i,j}$ by minimizing the KL divergence, which can be formulated as follows:

$$\mathcal{L}_{sdt}^{i2t} = \frac{1}{N} \sum_{i=1}^{N} \sum_{j=1}^{N} p_{i,j} \log \frac{p_{i,j}}{\alpha q_{i,j} + (1-\alpha)y_{i,j} + \epsilon}, \quad (18)$$

where $y_{i,j}$ is a true matching label, $y_{i,j} = 1$ means that $(f_i^{rgb,s}, f_i^{t,s})$ is a matched pair from the same identity. $\alpha$ is a parameter which controls the strength of regularization (a higher $\alpha$ leads to weaker regularization), while $\epsilon$ is a small number to avoid numerical problems. $q_{i,j}$ is treated as a constant and does not participate in backpropagation of $\mathcal{L}_{sdt}^{i2t}$. Symmetrically, the SDT loss $\mathcal{L}_{sdt}^{t2i}$ from text to image can be constructed by exchanging $f_i^{rgb,s}$ and $f_i^{t,s}$ in Eq.16 and by exchanging $f_i^{rgb,c}$ and $f_i^{t,c}$ in Eq.17. The bi-directional SDT loss is then calculated as follows:

$$\mathcal{L}_{sdt} = \mathcal{L}_{sdt}^{i2t} + \mathcal{L}_{sdt}^{t2i}. \quad (19)$$

During training, we employ the SDL to align multi-modal image and text features while decoupling color semantics. Meanwhile, the SDT loss is used to transfer color semantics knowledge to unseen retrieval task. The overall loss function is defined as follows:

$$\mathcal{L}_{total} = \mathcal{L}_{sdl} + \mathcal{L}_{sdt}. \quad (20)$$

## 4. Expriments

**Datasets.** Based on existing VI-ReID and omni multi-modal ReID datasets (Wu et al., 2017; Zhang & Wang, 2023; Zuo et al., 2026), we manually annotated and constructed the **SYSU-TVI** dataset, while collecting and organizing the **LLCM-TVI** dataset for TVI-ReID task. **SYSU-TVI** uses all image data from SYSU-MM01 (Wu et al., 2017) and includes manually annotated textual descriptions, containing 29,003 visible images, 15,712 infrared images and 44,715 textual descriptions. The text annotation process for SYSU-TVI consists of three steps: *I)* we first **manually** created fine-grained visible-light textual descriptions for each pedestrian based on the visible images. *II)* Then **MLLM** was used to perform text data augmentation in combination with images, enhancing the diversity of pedestrian descriptions, as presented in **Appendix A**. *III)* Finally, for each image, we selected a corresponding textual description for each image and **manually** corrected noisy information in augmented text. Notably, as presented in **Appendix B**, the textual annotations in SYSU-TVI emphasize a large amount of color-irrelevant information (e.g., positions of accessories, items not carried) to support a broader range of real-world situations, which traditional TI-ReID datasets (Li et al., 2017; Ding et al., 2021; Zhu et al., 2021; Zuo et al., 2026) overlooked. **LLCM-TVI** is constructed by selecting tri-modal data from the LLCM (Zhang & Wang, 2023) subset of the omini multi-modal ReID dataset ORBench (Zuo et al., 2026), containing 16,080 visible-light images, 13,171 infrared images and 16,080 textual descriptions.

**Evaluation Protocols.** Following previous work, we used **Rank-k** and mean Average Precision (**mAP**) to evaluate the model's retrieval performance across different modalities. The selection criterion for the best epoch is based on the average Rank-1 accuracy across two retrieval tasks. A higher average Rank-1 value indicates a better joint retrieval performance of the model.

**Implementation Overview.** Our work adopts CLIP's text and image encoders as the backbone (Radford et al., 2021; Qin et al., 2024) and initializes the network with pretrained weights (Jiang et al., 2025). During training, each batch contained mixed visible/infrared images from different identities, along with their corresponding textual descriptions. All details are described in **Appendix C**.

### 4.1. Comparison with the State-of-the-art Methods

To validate the effectiveness of our method, we compared it with various state-of-the-art approaches on SYSU-TVI and LLCM-TVI, as shown in Tab. 1 and Tab. 2.

**Performance on SYSU-TVI dataset.** As shown in Tab. 1, our method significantly improves the Rank-1 accuracy for both text-to-infrared and text-to-visible retrieval, achieving **66.21%** in text-to-infrared and **96.13%** in text-to-visible retrieval tasks, respectively, compared to the current best results 59.06% and 95.45% from RDE (Qin et al., 2024) + HAM (Jiang et al., 2025). In terms of mAP, our method achieves **47.63%** and **88.19%** for text-to-infrared and text-to-visible retrieval, respectively, outperforming the best results from RDE (Qin et al., 2024) + HAM (Jiang et al., 2025), which are 44.82% and 87.10%.

*Table 1.* Comparison with state-of-the-art methods on SYSU-TVI under our setting. It mainly evaluates Rank-1 accuracy (%), Rank-5 accuracy(%), Rank-10 accuracy (%), and mAP (%) for both text-to-infrared and text-to-visible image retrieval. It can be seen that our method achieves significant performance improvements on the both text-to-infrared and text-to-visible retrieval tasks.

| Methods | Venue | Backbone | T→IR | | | | T→RGB | | | |
|---|---|---|---|---|---|---|---|---|---|---|
| | | | R1 | R5 | R10 | mAP | R1 | R5 | R10 | mAP |
| IVT (Shu et al., 2022) | ECCVW22 | ViT-B+BERT | 22.09 | 44.71 | 55.37 | 19.58 | 65.79 | 78.63 | 83.59 | 56.13 |
| CFine (Yan et al., 2023) | TIP23 | CLIP+BERT | 29.61 | 48.78 | 71.39 | - | 78.33 | 89.95 | 97.44 | - |
| IRRA (Jiang & Ye, 2023) | CVPR23 | CLIP | 41.47 | 63.02 | 71.26 | 30.66 | 88.34 | 94.24 | 95.65 | 74.99 |
| TBPS-CLIP (Cao et al., 2024) | AAAI24 | CLIP | 27.95 | 45.70 | 54.90 | 20.02 | 75.16 | 89.33 | 92.80 | 61.18 |
| PLIP (Zuo et al., 2024) | NIPS24 | Swin-B+BERT | 18.41 | 32.02 | 41.49 | 12.39 | 87.08 | 93.63 | 95.66 | 68.52 |
| PLIP+SDM (Zuo et al., 2024) | NIPS24 | Swin-B+BERT | 41.97 | 59.95 | 69.31 | 25.08 | 91.88 | 94.49 | 96.76 | 77.85 |
| MLLM4Text-ReID (Tan et al., 2024) | CVPR24 | CLIP | 22.67 | 41.49 | 53.38 | 11.25 | 88.81 | 94.88 | 96.47 | 66.05 |
| RDE (Qin et al., 2024) | CVPR24 | CLIP | 52.20 | 69.39 | 78.18 | 38.10 | 90.91 | 94.35 | 95.68 | 74.52 |
| HAM (Jiang et al., 2025) | CVPR25 | CLIP | 31.05 | 47.94 | 58.40 | 17.55 | 95.66 | 97.48 | **98.81** | 81.02 |
| RDE+MLLM4Text-ReID (Tan et al., 2024) | CVPR24 | CLIP | 54.56 | 72.44 | 80.39 | 41.41 | 93.86 | 95.89 | 97.01 | 79.77 |
| RDE+HAM (Jiang et al., 2025) | CVPR25 | CLIP | 59.06 | 75.28 | 83.01 | 44.82 | 95.45 | 97.67 | 98.15 | 87.10 |
| **Ours** | - | CLIP | **66.21** | **81.28** | **87.25** | **47.63** | **96.13** | **97.92** | 98.36 | **88.19** |

*Table 2.* Comparison with state-of-the-art methods on LLCM-TVI under our setting.

| Methods | Venue | Backbone | T→IR | | | | T→RGB | | | |
|---|---|---|---|---|---|---|---|---|---|---|
| | | | R1 | R5 | R10 | mAP | R1 | R5 | R10 | mAP |
| CFine (Yan et al., 2023) | TIP23 | CLIP+BERT | 12.01 | 24.45 | 32.89 | - | 33.79 | 49.60 | 57.50 | - |
| IRRA (Jiang & Ye, 2023) | CVPR23 | CLIP | 19.46 | 35.19 | 44.23 | 15.06 | 44.53 | 61.88 | 69.46 | 34.75 |
| RDE (Qin et al., 2024) | CVPR24 | CLIP | 18.54 | 32.35 | 41.10 | 14.21 | 44.28 | 56.89 | 63.79 | 33.98 |
| PLIP (Zuo et al., 2024) | NIPS24 | Swin-B+BERT | 5.93 | 14.85 | 22.58 | 3.80 | 25.45 | 41.38 | 51.04 | 17.30 |
| MLLM4Text-ReID (Tan et al., 2024) | CVPR24 | CLIP | 13.76 | 24.93 | 30.78 | 10.03 | 36.49 | 52.33 | 59.21 | 25.71 |
| HAM (Jiang et al., 2025) | CVPR25 | CLIP | 23.42 | 36.31 | 43.36 | 16.76 | 59.22 | 72.97 | 78.91 | 43.55 |
| RDE+HAM (Jiang et al., 2025) | CVPR25 | CLIP | 34.59 | 50.23 | 58.16 | 28.18 | 63.44 | 74.76 | 79.94 | 53.50 |
| **Ours** | - | CLIP | **36.72** | **53.97** | **61.98** | **29.09** | **65.57** | **78.24** | **83.17** | **55.37** |

**Performance on LLCM-TVI dataset.** As shown in Tab. 2, our method also significantly improves the Rank-1 accuracy on LLCM-TVI, achieving **36.72%** and **65.57%** in text–infrared and text–visible image retrieval tasks, respectively, compared to the current best results from RDE (Qin et al., 2024) + HAM (Jiang et al., 2025), which are 34.59% and 63.44%. In terms of mAP, our method also reaches **29.09%** and **55.37%** for text-to-infrared and text-to-visible retrieval, surpassing the best results from RDE (Qin et al., 2024) + HAM (Jiang et al., 2025), which are 28.18% and 53.50%.

### 4.2. Ablation Study

To evaluate the contribution of each component, we conducted ablation studies on SYSU-TVI and LLCM-TVI, as shown in Tab. 3. We take RDE (Qin et al., 2024) without any additional components as the baseline (No.0), and assess the impact of each component by incrementally adding them. No. 0–2 and No. 3–5 denote settings with and without initialization using HAM (Jiang et al., 2025) pretrained weight, respectively.

**Effect of Semantic Decoupling Learning.** As shown in Tab. 3, we conducted experiments with and without the pretrained weight of HAM (Jiang et al., 2025). Compared to

No.0 and No.3, the proposed method achieves **3.10%** (No.1) and **4.28%** (No.4) Rank-1 improvements of text-to-infrared retrieval on SYSU-TVI, respectively, while maintaining the high Rank-1 of text-to-visible retrieval task. The same trend is also observed on LLCM-TVI, as shown in Tab. 3.

To further analyze the contribution of each component in SDL, we conduct ablation experiments on SYSU-TVI, as illustrated in Tab. 4, where No.0 denotes the original SDL module, No.1 refers to random replacement and masking of color words in the text training data for text-IR alignment, No.2 removes the channel augmentation (CA) component and No.3 excludes $\mathcal{L}_{orth}$. The experimental results demonstrate that both CA and $\mathcal{L}_{orth}$ improve module performance to varying degrees. Furthermore, the results of No.1 reveal that color variations have a minor impact on retrieval performance, which demonstrates the effectiveness of SDL in decoupling color attributes between text and images.

The experimental results demonstrate that SDL effectively extracts identity-related information for text-to-infrared and text-to-visible retrieval by aligning the decoupled color-related and color-irrelevant multi-modal representations separately.

**Effect of Semantic Distribution Transfer.** As shown in Tab. 3, compared to No.1 and No.4, the proposed method

*Table 3.* Ablation study about each component on SYSU-TVI and LLCM-TVI under our setting. HAM* denotes initializing with TI-ReID pretrained weights (Jiang et al., 2025).

| | Components | | | SYSU-TVI | | | | | | LLCM-TVI | | | | | |
| | | | | T→IR | | | T→RGB | | | T→IR | | | T→RGB | | |
| No. | HAM* | SDL | SDT | R1 | R10 | mAP | R1 | R10 | mAP | R1 | R10 | mAP | R1 | R10 | mAP |
|---|---|---|---|---|---|---|---|---|---|---|---|---|---|---|---|
| #0 | | | | 52.20 | 78.18 | 38.10 | 90.91 | 95.68 | 74.52 | 18.72 | 39.41 | 13.71 | 44.28 | 63.79 | 33.98 |
| #1 | | ✓ | | 55.30 | 80.94 | 39.46 | 89.77 | 96.56 | 75.31 | 20.84 | 47.26 | 16.33 | 45.04 | 68.77 | 34.14 |
| #2 | | ✓ | ✓ | 57.14 | 79.62 | 39.52 | 91.20 | 96.35 | 75.84 | 21.79 | 46.69 | 17.84 | 46.19 | 71.17 | 36.53 |
| #3 | ✓ | | | 59.06 | 83.01 | 44.82 | 95.45 | 98.15 | 87.10 | 34.59 | 58.16 | 28.18 | 63.44 | 79.94 | 53.50 |
| #4 | ✓ | ✓ | | 63.34 | 85.17 | 46.25 | 95.70 | 98.24 | 87.66 | 34.91 | 60.22 | 28.55 | 64.93 | 80.53 | 55.27 |
| #5 | ✓ | ✓ | ✓ | **66.21** | **87.25** | **47.63** | **96.13** | **98.36** | **88.19** | **36.72** | **61.98** | **29.09** | **65.57** | **83.17** | **55.37** |

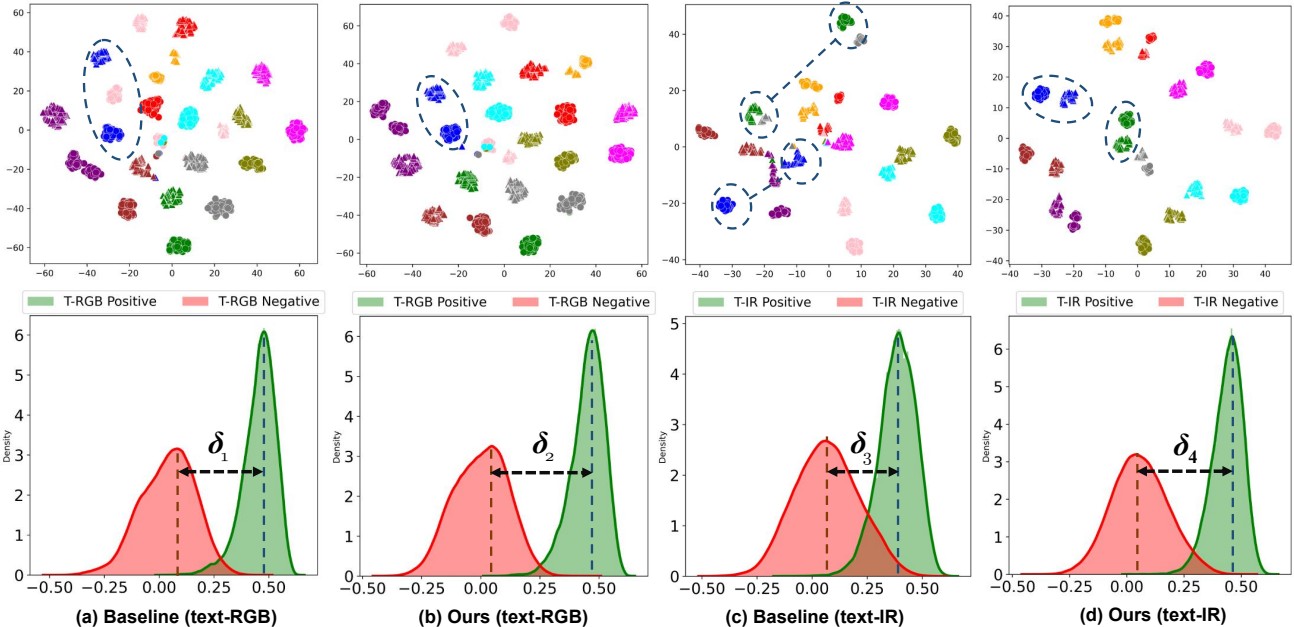

*Figure 3.* Visualization results of feature distributions and similarity distributions of different identities.

*Table 4.* Ablation study about each component of SDL on SYSU-TVI dataset.

| | | T→IR | | T→RGB | |
| No. | Components | R1 | mAP | R1 | mAP |
|---|---|---|---|---|---|
| #0 | SDL(ours) | 63.34 | 46.25 | 95.70 | 87.66 |
| #1 | + w/ color-operation | 63.03 | 46.57 | 95.96 | 85.75 |
| #2 | + w/o CA-image | 60.98 | 44.07 | 95.47 | 86.47 |
| #3 | + w/o $\mathcal{L}_{orth}$ | 61.50 | 41.97 | 95.34 | 84.23 |

achieves **1.84%** (No.2) and **2.87%** (No.5) Rank-1 improvements of text-to-infrared retrieval and **1.43%** (No.2) and **0.43%** (No.5) Rank-1 improvements of text-to-visible retrieval on SYSU-TVI, respectively. The same trend is also observed on LLCM-TVI, as shown in Tab. 3. The experimental results demonstrate that SDT effectively transfers pretrained knowledge to the downstream text-to-infrared retrieval task by aligning the similarity matrices of color-related and color-irrelevant multi-modal features.

## 4.3. Visualization

**Feature Distributions.** We project the features extracted from SYSU-TVI into the t-SNE (Maaten & Hinton, 2008) 2D space. As shown in Fig. 3, the t-SNE feature distribution demonstrates that our method significantly improves the discriminative ability for distinguishing different identities, while reducing extreme outliers of the same identity and samples with excessive cross-modal discrepancies. Meanwhile, Fig. 4 demonstrates that CSDT can effectively construct color-relevant and color-irrelevant feature subspaces, and reduce mutual interference during the alignment of different modality combinations.

**Similarity Distributions.** For the feature similarity distributions shown in Fig. 3 (corresponding to the 2D t-SNE feature space), it can be observed that the inter-class and intra-class distance distributions become increasingly well separated (i.e., $\delta_4 - \delta_3$ and $\delta_2 - \delta_1$), and notably, the intra-class distances are also significantly increased.

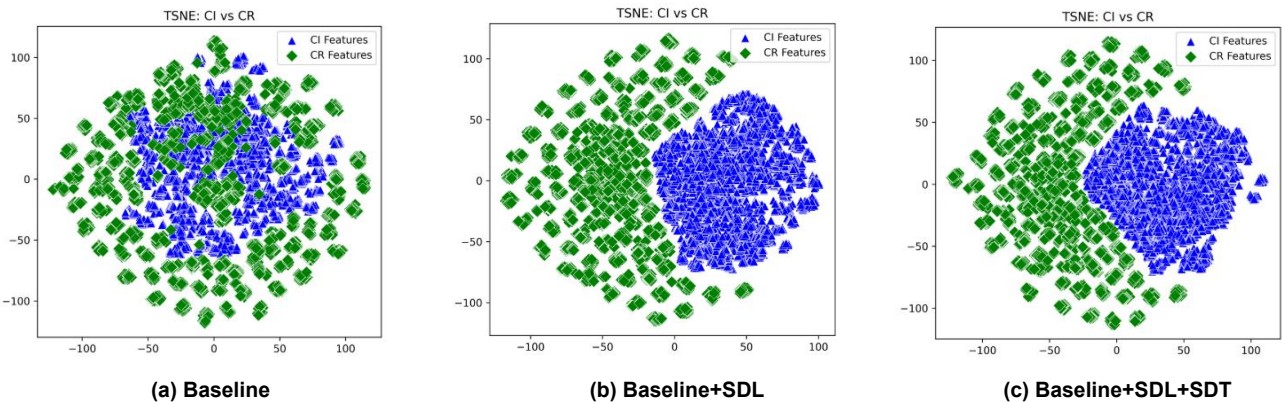

*Figure 4.* Visualization results of color-related and color-irrelevant feature distributions.

## 4.4. Parametric Analysis

To study the impact of different hyperparameter settings on performance, we conducted sensitivity analyses of the key hyperparameter $\alpha$ in Eq. 18 on SYSU-TVI. As shown in Fig. 5, too large $\alpha$ will cause the model to overfit the color-related distribution, leading to performance degradation, while too small $\alpha$ will cause $\mathcal{L}_{sdt}$ to degenerate into SDM loss (Jiang & Ye, 2023), thereby gradually reducing the improvement brought by SDT. Therefore, we set $\alpha = 0.9$ in all our experiments.

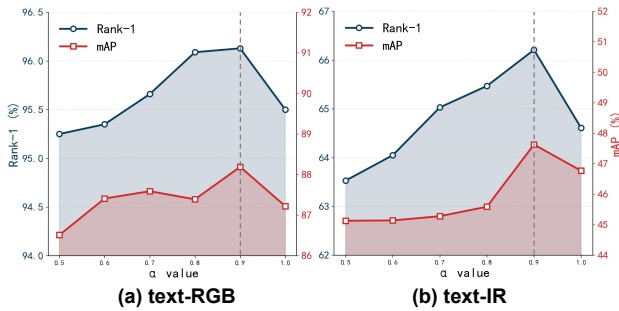

*Figure 5.* Variation of performance with different $\alpha$.

## 5. Conclusion

In this work, we introduce a novel task called TVI-ReID , and construct corresponding datasets, aiming to extend TI-ReID to nighttime and low-light conditions. Furthermore, we propose CSDT to perform cross-modal semantic decoupling and transfer for TVI-ReID. Semantic Decoupling Learning (SDL) is employed to align tri-modal features within two orthogonal feature subspaces, while the Semantic Distribution Transfer (SDT) facilitates the effective propagation of pretrained knowledge to the new downstream task. Experimental results demonstrate that our proposed framework outperforms current state-of-the-art methods.

## Acknowledgements

This work was supported by the National Natural Science Foundation of China under Grant (T2541022 and 62501428), the Postdoctoral Fellowship Program of China Postdoctoral Science Foundation under Grant (GZC20241268), the Hubei Provincial Natural Science Foundation of China under Grant (2025AFB219), and the Major Project of Science and Technology Innovation of Hubei Province under Grant (2025BEA002). The numerical calculations were supported by the supercomputing system at the Supercomputing Center of Wuhan University.

## Impact Statement

The proposed TVI-ReID framework has broad application potential in scenarios such as surveillance and personnel localization, contributing to public safety and social stability. However, this work may also raise ethical concerns related to privacy and surveillance, and thus should be deployed responsibly under transparent governance and regulatory oversight.

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

## A. Prompt Template for Dataset Construction

We present the prompt template for text data augmentation, as shown in Fig 6. This prompt, together with a pedestrian image from a random viewpoint and its corresponding manually annotated text description, was fed into the MLLM to generate pedestrian text descriptions with diverse viewpoints and linguistic styles.

---

**Prompt Template**

{{single pedestrain image}}
You are an annotator for image-text datasets.
We aim to train a text-to-infrared image person re-identification model, which requires textual descriptions for a batch of images.
Basic descriptions for the images are already available, and you need to enrich the human-related details according to the corresponding images. You will be given a basic description and an image. Keep all information from the original description, and revise it based on the subject's actions, postures, shooting angles and other visual cues. If glasses or clothing patterns mentioned in the basic description are occluded in the image, remove the relevant content.
While preserving all original information, appropriately adjust sentence structures, word order and writing styles to simulate the linguistic characteristics of different annotators.
Only output the revised description without any extra words before or after it.
{{few_shot_example}}
Basic description: {{basic description}}
Answer:

---

*Figure 6.* Prompt template for dataset construction.

## B. Details of SYSU-TVI Dataset

In this section, we present statistics and several tri-modal data samples from SYSU-TVI, as shown in Tab. 5 and Fig. 7. Notably, our fine-grained textual descriptions explicitly capture both **color-related** (e.g., clothing color) and **color-irrelevant** (e.g., positions of accessories, items not carried) identity-discriminative details to support a broader range of real-world situations.

*Table 5.* Dataset statistics.

| #Task | Train set | | | Test set | | |
|---|---|---|---|---|---|---|
| | #ID | #Image | #Text | #ID | #Image | #Text |
| text-RGB | 395 | 22258 | 22258 | 96 | 6775 | 6775 |
| text-IR | 395 | 11909 | 11909 | 96 | 3803 | 3803 |
| all | 395 | 34167 | 34167 | 96 | 10578 | 10578 |

## C. Implementation Details

Our work adopts CLIP's text and image encoders as the backbone (Radford et al., 2021; Qin et al., 2024) and initializes the network with pretrained weights (Jiang et al., 2025) to fully leverage the knowledge learned from large-scale datasets. Meanwhile, at the beginning of each epoch, the model divides the training pairs into clean and noisy samples based on their loss values, and only clean pairs are used for optimization in the current epoch, thus mitigating the adverse impact

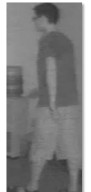 A slender man with short hair and glasses wears a gray short-sleeve T-shirt and khaki shorts that reach just past the knee, featuring multiple pockets. His bare calves are in strappy sandals with no socks, and he has a watch on his left wrist.

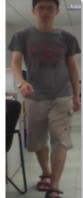 A slender man with short hair and glasses was wearing a gray short-sleeved T-shirt and khaki shorts with multiple pockets. He wore sandals with two straps on each foot, no socks, and a watch on his right wrist.

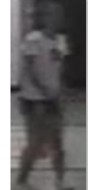 A girl with a long ponytail and a smooth forehead without bangs wears glasses. She has on a white short-sleeved T-shirt with a design and denim shorts. On her feet are yellow sneakers.

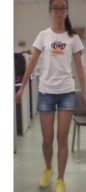 The girl with a long ponytail looked clean and simple in a white short-sleeved T-shirt and denim shorts, standing in bright yellow sneakers with a relaxed and natural posture.

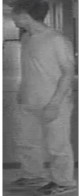 A slightly muscular man with short hair and no glasses wears a simple blue short-sleeve T-shirt without any patterns, highlighting his strong arms. He has on a pair of denim jeans and deep brown platform shoes.

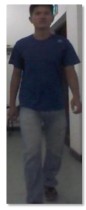 A sturdy male with short hair and no glasses, wearing a plain blue short-sleeved T-shirt that reveals toned muscles, jeans, and brown shoes, walked steadily and powerfully forward.

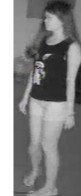 A lady with long hair, not wearing glasses, is confidently walking forward with a black sleeveless T-shirt, white cartoon rabbit pattern, white short pants, white sandals, a black backpack, and a light green bracelet on her right wrist.

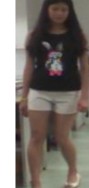 A long-haired woman without glasses walked in casual fashion. She wore a black sleeveless t-shirt with a white cartoon rabbit pattern on it, paired with slim white shorts and white sandals. She carried a black backpack on her back and wore a light green bracelet on her right wrist.

*Figure 7.* Visualization of the data samples from our SYSU-TVI Dataset.

of noise correspondence on cross-modal training (Qin et al., 2024). All training was conducted in PyTorch (Paszke et al., 2019) utilizing a single NVIDIA RTX 4090 GPU. It takes about 22GB memory for training and about 2GB memory for testing. About 40 miniutes are needed for training on SYSU-TVI and LLCM-TVI. The input image resolution was set to 384×128, and basic data augmentation techniques including horizontal flipping, random cropping, and random erasing were applied. We used the Adam (Howard et al., 2017) for optimization. During training, the base learning rate was set to 5e-6 with pretrained weights and was set to 1e-5 without pretrained weights, and each batch contained 96 mixed visible/infrared images from different identities, along with their corresponding 96 textual descriptions.

