# OpenReview forum: "Cross-Modal Semantic Decoupling and Transfer for Text-to-Visible-Infrared Person Re-Identification"
_ICML.cc/2026/Conference — ICML 2026 regular_

### Official Review · Reviewer_jqLY · 2026-02-27

**Soundness:** 3
**Presentation:** 4
**Significance:** 3
**Originality:** 3
**Overall Recommendation:** 5
**Confidence:** 4

**Summary:**

Tradional Text-to-Image Person Re-Identification (TI-ReID) often fails in low-light conditions,  infrared cameras offer a feasible solutions because thet can capture can capture pedestrian contours and texture information. Thus, this paper proposes a new task named Text-to-Visible-Infrared Person ReIdentification (TVI-ReID) and constructs a new tri-modal dataset SYSU-TVI and organized another dataset LLCM-TVI to support the TVI-ReID task. To solve the TVI-ReID, which faces challenges of complex hybrid discrepancies and semantic inconsistency, Cross-Modal Semantic Decoupling and Transfer (CSDT) framework is proposed. Specifically, Semantic Decoupled Learning (SDL) is designed to reducethe cross-modal discrepancies between text and dual-spectral images. In addition,  Semantic Distribution Transfer (SDT) aims to transfers knowledge by minimizing the KL divergence between the text-image similarity distributions of color-related and color-irrelevant domains. Extensive experiments on SYSU-TVI and LLCM-TVI demonstrate the effectiveness of the proposed CSDT framework.

**Compliance With Llm Reviewing Policy:**

Affirmed.

**Final Justification:**

The authors have thoroughly addressed my concerns with reasonable and complete experiments. The work is solid and meaningful to the community and my final recommendation is Accept.

**Key Questions For Authors:**

1. Could you provide further justification or experimental evidence for decoupling semantics into color-related and color-irrelevant features via channel splitting.

2. How sensitive is the model to the hyperparameter.

3. Could you elaborate on the dataset construction process, particularly the quality assurance of textual annotations.

**Limitations:**

More failure cases and scope of applicability would be helpful to understand the limitations of the proposed methods.

**Strengths And Weaknesses:**

Strengths:

1.The paper presents a clear and logical structure, effectively motivating the research from the perspectives of task definition, methodology, and experimental validation. The introduction of the new TVI-ReID task and the corresponding datasets (SYSU-TVI and LLCM-TVI) addresses a practical and underexplored problem in 24-hour surveillance, which holds significant real-world applicability.

2.The proposed CSDT framework is well-designed. The Semantic Decoupling Learning (SDL) module reasonably distinguishes between color-related and color-irrelevant identity cues across RGB and infrared modalities, while the Semantic Distribution Transfer (SDT) strategy effectively leverages pretrained knowledge for the text-to-infrared retrieval task. Ablation studies on both datasets validate the individual contributions of SDL and SDT.

3.The proposed method achieves state-of-the-art performance on the newly constructed datasets, demonstrating its effectiveness in handling the TVI-ReID task.

Weaknesses:

1.A key concern is whether the explicit decomposition of color-related and color-irrelevant cues via channel splitting in SDL is fundamentally justified or remains a heuristic. Given that the modality gap between RGB and infrared images involves more than just color information (e.g., texture, contrast, and thermal patterns), the design may oversimplify the problem. It would be helpful to see additional experiments or analysis that justify the rationality of this decoupling.

2.The ablation study could be more comprehensive. In particular, the impact of the orthogonal loss L_{orth} in Eq. (14) is not individually analyzed. It would strengthen the paper to isolate and evaluate its contribution to the overall performance.

3.The hyperparameter α in Eq. (17) appears to have a significant influence on  final retrieval results. The current dependence on empirical tuning may limit the generalizability of the method across different datasets or settings.

4.While the datasets are a contribution, more details on the annotation process and quality control would help assess their reliability.

---

> ### Author Rebuttal · Authors · 2026-03-30
>
> We thank Reviewer jqLY for the positive feedback on the task contribution, framework design, and performance. We hope that this rebuttal can address your concerns regarding theoretical explanation of SDL, effect of orthogonal loss and hyperparameter and reliability of dataset construction.
>
> ***Q1: Theoretical Explanation of SDL (Key Question 1)***
>
> The purpose of introducing **channel augmentation** [1] is to construct paired color-related and color-irrelevant features at the image-level for RGB image. For the necessity and rationality of this operation, as well as supporting experimental evidence, please refer to our response to Q4 from reviewer ifeT and Q2 from reviewer uFie.
>
> In the field of person re-identification, orthogonal loss has been used for feature-level information disentanglement [3]. The underlying idea is that metric learning mainly relies on pulling apart and pushing together feature distances to build a discriminative feature space, while our orthogonal loss reduces the interference caused by such distance variations between two subspaces by enforcing orthogonal hyperplanes.
>
> In summary, the two key designs of SDL have been well validated in other ReID tasks [1] [2] [3], and our additional ablation experiments (please see our response to Q2 bellow and response to Q4 from reviewer ifeT) further confirm their critical roles.
>
> ***Q2: Effect of Orthogonal Loss $\mathcal{L}\_{orth}$ (Weakness 2)***
>
> The ablation results for the orthogonal loss are presented as follows. Experiments demonstrate the positive effect of the orthogonal loss on retrieval performance.
>
> | Method                   | R1 (t2ir) | mAP (t2ir) | R1 (t2rgb) | mAP (t2rgb) |
> | ------------------------ | --------- | ---------- | ---------- | ----------- |
> | w/o $\mathcal{L}_{orth}$ | 61.50     | 41.97      | 95.34      | 84.23       |
> | w/ $\mathcal{L}_{orth}$  | 63.34     | 46.25      | 95.70      | 87.66       |
>
> ***Q3: Effect of Hyperparameter $\alpha$ (Key Question 2)***
>
> The discussion of the hyperparameter $\alpha$ has already been analyzed in **Section 4.4**. When $\alpha$ approaches 0, the loss degenerates into the SDM loss, simply aligning the text modality with the single-channel image modality. When $\alpha$ approaches 1, the model excessively fits the color-related distribution. Overall, for most values of $\alpha$, the SDT module **improves performance**, and the size of the improvement shows a clear pattern as $\alpha$ changes.
>
> ***Q4: Dataset Construction (Key Question 3)***
>
> We provide a detailed description of the annotation pipeline for SYSU-TVI as follows:
>
> **Step 1.** Eleven annotators manually labeled fine-grained textual descriptions for each identity. During annotation, images from multiple viewpoints were jointly considered. The types of attributes included in the descriptions were strictly controlled to ensure both the quality and correctness of the annotations.
>
> **Step 2.** We introduce an MLLM to perform data augmentation by taking randomly sampled viewpoint images together with the manually annotated descriptions from Step 1 as input. The motivation is twofold:
>
> (1) different viewpoints of the same person may involve the appearance or disappearance of key items in text descriptions.
>
> (2) the MLLM can generate diverse annotation style while preserving correctness.
>
> Applications of MLLMs in multimodal ReID dataset construction can be found in prior work [4] and [5].
>
> During the MLLM generation process, we explicitly constrain the model to follow the manually annotated descriptions from Step 1, allowing only modifications related to viewpoint changes or variations in language style. More **implementation details (e.g., prompts)** will be provided in next version of paper for reproducibility.
>
> **Step 3.** With the assistance of the same eleven annotators, we further verified and refined all text–image pairs to eliminate potential hallucinations and ensure consistency.
>
> [1] Ye M, Wu Z, Chen C, et al. Channel augmentation for visible-infrared re-identification[J]. IEEE Transactions on Pattern Analysis and Machine Intelligence, 2023, 46(4): 2299-2315.
>
> [2] Hu Z, Yang B, Ye M. Empowering visible-infrared person re-identification with large foundation models[J]. Advances in Neural Information Processing Systems, 2024, 37: 117363-117387.
>
> [3] Zhang Q, Wang L, Patel V M, et al. View-decoupled transformer for person re-identification under aerial-ground camera network[C]//Proceedings of the IEEE/CVF Conference on Computer Vision and Pattern Recognition. 2024: 22000-22009.
>
> [4] Tan W, Ding C, Jiang J, et al. Harnessing the power of mllms for transferable text-to-image person reid[C]//Proceedings of the IEEE/CVF Conference on Computer Vision and Pattern Recognition. 2024: 17127-17137.
>
> [5] Jiang J, Ding C, Tan W, et al. Modeling thousands of human annotators for generalizable text-to-image person re-identification[C]//Proceedings of the Computer Vision and Pattern Recognition Conference. 2025: 9220-9230.

---

> > ### Author Rebuttal · Reviewer_jqLY · 2026-04-04
> >
> > My concerns have been addressed. Thank you to the reviewers for the clarification, and I will raise my score.

---

> > > ### Author Response · Authors · 2026-04-04
> > >
> > > We sincerely thank Reviewer jqLY for the encouraging and positive feedback. We are also pleased that our responses have addressed your concerns. The professional suggestions have enabled us to better refine important technical details. We will incorporate these insights into the revision to further strengthen the clarity and completeness of our paper.
> > >
> > > Thank you again for your time and valuable comments!

---

### Official Review · Reviewer_ifeT · 2026-03-01

**Soundness:** 3
**Presentation:** 3
**Significance:** 4
**Originality:** 4
**Overall Recommendation:** 5
**Confidence:** 3

**Summary:**

To address the performance limitations of traditional Text-to-Image Person Re-Identification (TI-ReID) in nighttime or low-light environments, this paper proposes the Text-to-Visible-Infrared Person Re-Identification (TVI-ReID) task. To tackle the core challenges—namely the complex modality gaps among the three modalities (text, visible, and infrared) and the semantic inconsistency of pre-trained models (e.g., CLIP) in downstream text-infrared matching—this work introduce the CSDT framework: (1) Semantic Decoupling Learning (SDL) constructs color-related and color-irrelevant feature subspaces. It extracts features through two orthogonal branches and incorporates an Orthogonal Loss to minimize interference between them. (2) Semantic Distribution Transfer (SDT) transfers the knowledge of "text-visible" alignment from the pre-trained model to the "text-infrared" alignment task by minimizing KL divergence, thereby guiding the learning of the color-irrelevant space. The proposed method also introduces two new tri-modal datasets, SYSU-TVI and LLCM-TVI, and significantly outperforms existing state-of-the-art methods in terms of performance.

**Compliance With Llm Reviewing Policy:**

Affirmed.

**Final Justification:**

The authors have adequately addressed my concerns and I am happy to maintain my positive rating.

**Key Questions For Authors:**

1. There appears to be a potential notation error in Figure 2 concerning the SDL module. According to the definitions in Equations (7)
and (8), $f_{i}^{t,c}$ and $f_{i}^{t,s}$ represent color-related and color-irrelevant textual features, respectively. However, in the
diagram, $f_{i}^{t,s}$ is associated with the Color-related branch, while $f_{i}^{t,c}$ is linked to the Color-irrelevant branch. Could
the authors clarify if these labels are inadvertently swapped?
2. Based on the dataset description, the framework does not seem to construct a separate set of text descriptions devoid of color
information for the color-irrelevant branch. Can the authors confirm whether this branch still extracts features from the original full
text containing explicit color attributes (e.g., "red", "blue") across all experiments?
3. Following Question 2, since the method does not explicitly model the concept of "color" itself, how can it be proven that the
decoupled branches are specifically capturing color information rather than other latent features? Would it be beneficial to include an
ablation study—such as comparing features extracted from original texts versus those from "color-word-removed" versions—to
quantitatively demonstrate that the color-irrelevant branch successfully filters out color-related semantics?
4. The paper employs grayscale images to simulate the infrared modality for assisting the learning of color-irrelevant information.
What is the specific necessity of this intermediate step? If the grayscale augmentation were removed, and the model relied solely on
real IR and visible images for alignment, would the semantic decoupling of color-irrelevant information still be effectively achieved?

**Limitations:**

Please refer to the key question.

**Strengths And Weaknesses:**

**Strengths**:
1. This paper introduces TVI-ReID, a novel task that aligns better with the requirements of 24-hour surveillance. It successfully fills the research gap in text-based cross-modal retrieval within the infrared domain.
2. The authors astutely identify the attribute discrepancy of "color" between visible and infrared modalities. The proposed SDL module effectively achieves semantic separation, addressing a core challenge in tri-modal alignment.
3. Extensive Experimental Validation: The method has been compared against the latest SOTA methods on multiple datasets, demonstrating significant improvements in both Rank-1 and mAP metrics.

**Weaknesses**:
1. The analysis regarding the necessity of introducing grayscale images is not sufficiently detailed. It remains unclear how much this step contributes to the final performance compared to using raw infrared data.
2. There is a lack of sufficient quantitative or qualitative analysis to substantiate that the method truly achieves effective decoupling of color information as claimed.

---

> ### Author Rebuttal · Authors · 2026-03-30
>
> We thank Reviewer ifeT for the positive feedback on the application value, dataset contribution, and the clarity of our presentation. We hope that this rebuttal can address your concerns regarding annotation error, effect of color words, and necessity of channel augmentation images.
>
> ***Q1: Annotation Error (Key Question1)***
>
> We sincerely thank you for carefully identifying this mistake. We will promptly correct this annotation error in the next revision.
>
> ***Q2: Input of Color-irrelevant Branch of $F^T$ (Key Question2)***
>
> We would like to clarify that the color-irrelevant branch of $F^T$ still extracts features from the original full text containing color attributes.
>
> ***Q3: Effect of Color Words in Color-irrelevant Branch (Key Question3)***
>
> For color-irrelevant features in the text modality, although these features are extracted from the original textual descriptions, they only **performed contrastive learning with grayscale images** (i.e., single-channel channel-augmented images and infrared images) in the triplet loss of SDL. This suppresses the influence of color-related semantics during feature matching.
> We designed ablation experiments to analyze the effect of color words in SDL. We identified all color words in the input text for the color-invariant branch and applied masking and replacement operations. The experiments show that the absence of color words in the input has a **minimal impact** on performance, which proves that color-irrelevant branch successfully filters out color-related semantics.
>
> | Method   | R1 (t2ir) | mAP (t2ir) | R1 (t2rgb) | mAP (t2rgb) |
> | --------- | --------- | ---------- | ---------- | ----------- |
> | w/ operations-to-color | 63.03     | 46.57      | 95.96      | 85.75       |
> | w/o operations-to-color| 63.34     | 46.25      | 95.70      | 87.66       |
>
> ***Q4：Necessity of Channel Augmentation Images (Key Question4)***
>
> The necessity of channel augmentation images can be summarized as follows:
>
> (1) Both the SDL and SDT modules require **image-level paired** color-related and color-irrelevant data. However, due to the difficulty of collecting paired visible-infrared images, most of the existing VI-ReID datasets do not provide image-level paired RGB-IR data. Constructing paired color-irrelevant single-channel images from RGB images helps address this issue.
>
> (2) Since simultaneous participation of RGB and IR images in metric learning with text may lead to model confusion in text-IR alignment, the IR Proxy is a key component in SDL, used to independent the color-irrelevant features of RGB images and construct corresponding image data unrelated to textual color descriptions, which can be seen as an augmentation, facilitating joint metric learning of color-irrelevant features from both CA and IR images with text. In SDL loss, ​$f_i^{t,s}$ undergoes contrastive learning with both $f_i^{rgb,s}$ and $f_i^{ir,s}$ to establish a stable image-text mapping and eliminate the interference caused by color information.
>
> (3) We removed the introduction of channel augmentation images in SDL and conducted ablation experiments to demonstrate the necessity of such images in SDL. Experimental results show that channel augmentation has a crucial impact on the performance enhancement of SDL.
>
> | Method       | R1 (t2ir) | mAP (t2ir) | R1 (t2rgb) | mAP (t2rgb) |
> | ------------ | --------- | ---------- | ---------- | ----------- |
> | w/o CA-image | 60.98     | 44.07      | 95.47      | 86.47       |
> | w/ CA-image  | 63.34     | 46.25      | 95.70      | 87.66       |
>
> For further details on the justification of the channel augmentation operation, please refer to our response to Q2 from reviewer uFie.

---

> > ### Author Rebuttal · Reviewer_ifeT · 2026-04-02
> >
> > The authors have thoroughly addressed my concerns with reasonable and complete experiments.

---

> > > ### Author Response · Authors · 2026-04-02
> > >
> > > We sincerely thank Reviewer ifeT for the positive comments regarding our paper, which encourage us to pursue higher standards in our research and writing. We are also pleased that our responses have fully addressed your concerns. Your professional and insightful comments have helped us further clarify key technical details (e.g., annotation errors, and input of the color-irrelevant branch of $F^T$), providing valuable guidance for improving our final manuscript.
> > >
> > > Thank you once again for dedicating time and providing thoughtful comments!

---

### Official Review · Reviewer_5DBw · 2026-03-11

**Soundness:** 4
**Presentation:** 4
**Significance:** 4
**Originality:** 4
**Overall Recommendation:** 5
**Confidence:** 5

**Summary:**

This paper proposes a new task, TVI-ReID, which aims to retrieve visible and infrared images using textual descriptions of pedestrians, in order to address the limitation of traditional text–image ReID methods in nighttime scenarios. To tackle the complex hybrid discrepancies caused by dual cross-modal retrieval from three modalities, the authors introduce the SDL module. In addition, the SDT module is proposed to address the semantic inconsistency between pretraining and downstream tasks. Experimental results demonstrate effectiveness of proposed method.

**Compliance With Llm Reviewing Policy:**

Affirmed.

**Final Justification:**

My concerns have been adequately addressed with the detailed rebuttal. I would like to keep 5.

**Key Questions For Authors:**

Please see the weaknesses.

**Limitations:**

Yes

**Strengths And Weaknesses:**

Strengths
1.The proposed method and modules are not overly complex or redundant, and the design avoids stacking too much unrelated components, making it easy for future work to build upon.
3.The overall structure of the paper is well organized, clearly presenting the key motivation and problems to be addressed and the corresponding solutions, which makes the paper readable and easy to follow.
4.The designs proposed to address the two key challenges are intuitive and well motivated. The authors also provide experimental results that validate the effectiveness of these designs.
5.To support the proposed task, the authors annotate and construct a dedicated benchmark, which requires substantial manual effort. In addition, the paper evaluates multiple mainstream TI-ReID methods on the proposed benchmark, demonstrating a considerable amount of experimental work.


Weaknesses
1.The paper introduces a dual-stream encoding architecture. Would this design significantly increase the number of model parameters and lead to additional training costs? The authors should discuss this aspect.
2.In Section 3.1, the channel augmentation operation should be formally defined with one or two equations.
3.The Identity Cues mentioned in Figure 2 are rarely referenced in the main text. Do they correspond to the Identity-Related Features described elsewhere in the paper?
4.Could this modification prevent the full CLIP pre-trained weights from being loaded, or cause some existing CLIP-based TI-ReID pre-trained weights to be ineffective?
5.Regarding the hyperparameter alpha in Eq. 17, the paper describes its role as controlling the strength of regularization. However, according to Eq.17, the regularization term is actually scaled by (1 - alpha). The authors should consider revising the wording to accurately reflect this.

---

> ### Author Rebuttal · Authors · 2026-03-31
>
> We thank Reviewer 5DBw for the positive feedback of our work. We hope that this rebuttal can address your concerns regarding training cost,  formalization, and the description of the hyperparameter $\alpha$.
>
> ***Q1: Additional Training Cost and Pretrained Weights (Weakness 1 & Weakness 4)***
>
> The CSDT model has approximately 178M parameters in total. Compared with the original TI-ReID framework, which has around 153M parameters, the increase is only about 25M parameters, mainly in the embedding layer.
>
> During both training and inference, CSDT only selects the corresponding embedding branch to encode each image/text input. Therefore, the computational complexity remains the same as that of the original CLIP framework.
>
> When loading pretrained weights, we replicate the weights of the embedding layers for different shallow encoders to adapt to pretrained TI-ReID models (e.g., HAM, MLLM4Text-ReID).
>
> The inference time of RDE and CSDT on an RTX 4090 GPU is compared as follows:
>
> | Method | time (ms per image) | time (ms per text) |
> | ------ | ------------------- | ------------------ |
> | RDE    | 0.967               | 0.151              |
> | ours   | 0.873               | 0.137              |
>
> ***Q2: Formalization of Channel Augmentation (Weakness 2)***
>
> Following [1], we provide a formal definition of Channel Augmentation:
>
> $x^{v,R}_i=(x^R_i,x^R_i,x^R_i)$,
>
> $x^{v,G}_i=(x^G_i,x^G_i,x^G_i)$,
>
> $x^{v,B}_i=(x^B_i,x^B_i,x^B_i)$,
>
> $x^{v,CI}_i \sim \\{x^{v,R}_i,x^{v,G}_i,x^{v,B}_i\\}$.
>
> where $x^{v,R}_i$, $x^{v,G}_i$, and $x^{v,B}_i$ denote the single-channel images of R, G, and B, respectively, and $x^{v,CI}_i$ denotes the color-irrelevant image, $\sim$ denotes randomly sampling from a set.
> We will include a formal description of channel augmentation in the next version of the paper.
>
> [1] Ye M, Wu Z, Chen C, et al. Channel augmentation for visible-infrared re-identification[J]. IEEE Transactions on Pattern Analysis and Machine Intelligence, 2023, 46(4): 2299-2315.
>
> ***Q3:  Identity Cues vs Identity-Related Features (Weakness 3)***
>
> Identity Cues refer to directly observable discriminative signals, such as color, texture, and clothing attributes, which are relatively explicit.
>
> Identity-Related Features refer to learned feature representations associated with identity, which integrate multiple cues and are more abstract. We will include a brief discussion of these two concepts in the next version of the paper.
>
> ***Q4:  Hyperparameter $\alpha$ (Weakness 5)***
>
> We acknowledge that the description of the hyperparameter $\alpha$ is not ideal, and we will improve it in the next revision of the paper.

---

> > ### Author Rebuttal · Reviewer_5DBw · 2026-04-01
> >
> > My concerns have been adequately addressed with the detailed rebuttal and the experiments in the SUPP. I would like to keep 5. Thanks for the rebuttal.

---

> > > ### Author Response · Authors · 2026-04-02
> > >
> > > We sincerely thank Reviewer 5DBw for the constructive and positive feedback. We are also pleased that our responses fully resolved your concerns regarding training cost, formalization and presentation. These insightful suggestions encourage us to further enhance both the technical depth and overall quality of our paper. We will thoughtfully incorporate these discussions to further improve the final manuscript.
> > >
> > > Thank you once again for your support of the practical value and overall quality of our research!

---

### Official Review · Reviewer_uFie · 2026-03-12

**Soundness:** 3
**Presentation:** 3
**Significance:** 3
**Originality:** 3
**Overall Recommendation:** 4
**Confidence:** 4

**Summary:**

The paper proposes a novel task called Text-to-Visible-Infrared Person Re-Identification (TVI-ReID) to address the performance limitations of standard Text-to-Image ReID (TI-ReID) in nighttime or low-light scenarios. To tackle the hybrid discrepancies across text, visible, and infrared modalities, the authors introduce the Cross-Modal Semantic Decoupling and Transfer (CSDT) framework. This framework utilizes Semantic Decoupling Learning (SDL) to separate color-related and color-irrelevant features and Semantic Distribution Transfer (SDT) to adapt pretrained visible-text alignment knowledge to the text-infrared matching task. Additionally, the authors construct two tri-modal datasets, SYSU-TVI and LLCM-TVI, to benchmark the new task.

**Compliance With Llm Reviewing Policy:**

Affirmed.

**Final Justification:**

My concerns have been adequately addressed.

**Key Questions For Authors:**

The introduction of the TVI-ReID task is highly valuable for building robust 24-hour surveillance systems, and the dataset contributions are commendable. Furthermore, the proposed dual-branch decoupling strategy—leveraging single-channel augmentation to compel the network to separate semantics—represents a creative and methodologically interesting engineering solution to a complex multi-modal problem.

However, my recommendation is borderline due to several theoretical and experimental concerns that currently limit the paper's rigor. The authors are strongly encouraged to address the following critical points during the rebuttal phase:

1. Resolution of the SDL vs. SDT Conflict: The authors must provide theoretical justification or empirical evidence (e.g., feature visualizations or attention map comparisons) explaining why minimizing the KL divergence in SDT does not inadvertently force the color-irrelevant subspace ($\mathcal{F}^s$) to re-absorb color biases from the color-related subspace ($\mathcal{F}^c$), which would otherwise defeat the purpose of SDL.
2. Justification of the IR Proxy: Acknowledge the physical gap between single-channel RGB and thermal/NIR infrared. The authors should briefly clarify why channel-augmentation serves as an adequate and effective empirical proxy for IR in this specific ReID context, despite the underlying physical differences.
3. Verification of Baseline Fairness: Explicitly clarify the training configuration used for the baselines (e.g., IRRA, RDE, PLIP) presented in Tables 1 and 2. The authors must confirm whether these baselines were granted the exact same data exposure (i.e., trained on the mixed Visible and Infrared batches of SYSU-TVI and LLCM-TVI). If not, adjusted experimental results ensuring a strictly fair comparison are required.

**Limitations:**

Please refer to Weaknesses.

**Strengths And Weaknesses:**

**Strengths**:

1. Extending text-based person retrieval to 24-hour surveillance (handling both day/RGB and night/IR) is a practical engineering problem with real-world application value.

2. The annotation of textual descriptions for existing RGB-IR datasets (SYSU-MM01 and LLCM) contributes to the community, facilitating future research in this niche area.

3. The paper is generally well-structured and easy to follow.

**Weaknesses**:

1. Theoretical Consistency Between Semantic Decoupling (SDL) and Transfer (SDT) . The authors explicitly state that color is "identity-irrelevant and should be eliminated as interference in text-to-infrared retrieval". Consequently, they use an orthogonal loss (\mathcal{L}_{orth}) in the SDL module to strictly decouple the color-related subspace \mathcal{F}^c from the color-irrelevant subspace \mathcal{F}^s. However, in the SDT module, the authors compute a KL divergence loss (\mathcal{L}_{sdt}), forcing the color-irrelevant similarity distribution ($p_{i,j}$) to mimic the color-related similarity distribution ($q_{i,j}$). If a text query describes a "red shirt," the color-related similarity $q_{i,j}$ will be inherently high for a visible image of a person in a red shirt. By forcing $p_{i,j}$ to mimic $q_{i,j}$, the framework is explicitly forcing the "color-irrelevant" branch to replicate the metric space geometry of the "color-related" branch. This mathematically re-entangles the features and completely destroys the decoupling effort achieved by $\mathcal{L}_{orth}$. The authors fail to justify why minimizing this specific KL divergence does not simply collapse $\mathcal{F}^s$ back into encoding color biases.

2. Physical Discrepancy in the Infrared Proxy. To extract color-irrelevant features from visible images, the authors use "Channel-Augmentation" by randomly selecting a single channel (R, G, or B). While this serves as an empirical workaround, it makes a very strong assumption. A single RGB channel (e.g., the Red channel) is not merely "colorless"; its structural intensity is directly dictated by the original color (a red shirt appears extremely bright in the R channel and dark in the B channel). More importantly, single-channel RGB is an imperfect physical proxy for Infrared (Thermal/NIR). Infrared captures heat radiation or near-infrared reflectance, which behaves entirely differently from visible light reflectance. Treating $V_i^r$, $V_i^g$, or $V_i^b$ as a semantically equivalent proxy to $V_i^{ir}$ ignores the fundamental physical gap between these modalities.

3. The core premise of the proposed Semantic Decoupling Learning (SDL) relies heavily on the strict dichotomy that color information is crucial for visible (RGB) retrieval but entirely irrelevant and noisy for infrared (IR) retrieval. While it is generally true that IR images lack chromatic data, this binary decoupling overlooks nuanced scenarios where "color" semantics in text descriptions (e.g., "white shirt" vs. "black shirt") correlate strongly with luminance or texture patterns visible in the IR spectrum.

4. In real-world surveillance scenarios, user queries are often noisy, ambiguous, or contain contradictory information (e.g., "a person wearing a dark jacket" in an IR image where contrast is low, or a user mistakenly mentioning a color that doesn't exist in the IR spectrum). The paper lacks experiments evaluating the model's robustness against such natural language variations.

5. The core mechanism of semantic decoupled learning (SDL) is to divide the "color dependent" and "color independent" features into different subspaces. The purpose of this method is to reduce the noise in infrared retrieval, but its premise is that these feature sets are linearly separable and mutually exclusive. In fact, many discriminative identity clues under visible light are essentially related to color (for example, specific shadows on logos, gradients on shoes, or subtle skin color differences related to race or age). By strictly implementing the orthogonality or separation between these domains, the fine texture pattern correlation that appears at the same time as the color but is still visible in the brightness channel of the infrared image is discarded.

6. Baseline Fairness. To prove the superiority of the CSDT architecture, the authors must ensure a strictly fair comparison regarding data exposure. Standard VI-ReID models should be adapted to accept text inputs, or at the very least, the TI-ReID baselines must be re-trained using the exact same tri-modal dataset and modality-mixing strategies. Without this verification, the performance gap might simply reflect the CSDT model's exposure to IR data during training, rather than the architectural brilliance of the framework itself.

---

> ### Author Rebuttal · Authors · 2026-03-30
>
> We thank Reviewer uFie for the positive feedback on application value, dataset contribution, and clarity of presentation. We hope that this rebuttal can address your concerns regarding module conflicts, the infrared proxy, and experimental fairness.
>
> ***Q1: Theoretical Consistency Between SDL and SDT (Key Question 1 & Weakness 1)***
>
> We explain why SDL and SDT are not theoretically conflicting from three perspectives：
>
> (1) **Motivational perspective.**  The purpose of the color-irrelevant branch is to eliminate the interference caused by **non-authentic color information** in the **input data** (e.g., the gray-level appearance in infrared images), rather than to strictly enforce that the **labels** (i.e., soft label $q_{i,j}$ and hard label $y_{i,j}$) are independent of the **real clothing colors** of pedestrians. Therefore, the two modules are not in conflict at the level of motivation.
>
> (2) **Formulation perspective.**  The SDL loss enforces an orthogonality constraint $\mathcal{L}_{orth}$ between $\mathcal{F}^c$​ and $\mathcal{F}^s$​​. In SDT loss, the two distributions $q\_{i,j}$ and $p\_{i,j}$ are computed within their respective feature subspaces (i.e., $q\_{i,j}=\frac{exp(S(f\_i^{rgb,c},f\_j^{t,c})/\tau)}{\sum\_k exp(S(f\_i^{rgb,c},f\_k^{t,c})/\tau)}$ is computed from $f\_i^{rgb,c}$​ and $f\_i^{t,c}$​ within the hyperplane $\mathcal{F}^c$, while $p\_{i,j}=\frac{exp(S(f\_i^{rgb,s},f\_j^{t,s})/\tau)}{\sum_k exp(S(f\_i^{rgb,s},f\_k^{t,s})/\tau)}$ is computed from $f\_i^{rgb,s}$​ and  $f\_i^{t,s}$​ within hyperplane $\mathcal{F}^s$). KL divergence only affects the internal structure within each subspace and does not reduce the distance between the two hyperplanes, which is not in conflict with the optimization objective of SDL.
>
> (3) **Visualization**
> The visualization result is presented at https://anonymous.4open.science/r/ci-cr-tsne-3F1B/tsne-CI-CR.pdf. The result shows that incorporating SDT does not lead to feature space entanglement and does not offset the benefits of SDL.
>
>
> ***Q2: Justification of the IR Proxy (Key Question 2 & Weakness 2)***
>
> The justification can be summarized as follows:
>
> (1) It should be clarified that we not only align CA (IR Proxy)-Text but also align IR-Text. CA [1] can be seen as an augmentation: by simulating variations unrelated to textual color descriptions, it helps the model learn color-invariant feature representations. We do not ignore the fundamental physical gap between CA and IR.
>
> (2) In each training iteration, the channel selection from R,G,B for each image is fully randomized, ensuring that the resulting single-channel intensity does not maintain a stable correspondence with the color descriptions (**Weakness 2**).
>
> (3) In addition, both the SDL and SDT modules require **image-level paired** color-related and color-irrelevant data. However, due to the difficulty of collecting paired visible-infrared images, most of the existing VI-ReID datasets do not provide image-level paired RGB-IR data. Constructing paired color-irrelevant single-channel images from RGB images helps address this issue.
>
> (4)  Experimental results show that channel augmentation has a crucial impact on the performance enhancement of SDL.
>
> | Method       | R1 (t2ir) | mAP (t2ir) | R1 (t2rgb) | mAP (t2rgb) |
> | ------------ | --------- | ---------- | ---------- | ----------- |
> | w/o CA-image | 60.98     | 44.07      | 95.47      | 86.47       |
> | w/ CA-image  | 63.34     | 46.25      | 95.70      | 87.66       |
>
> ***Q3: Baseline Fairness (Key Question 3 & Weakness 6)***
>
> All non-pretrained baselines (i.e., IVT, CFine, IRRA, TBPS-CLIP, RDE) are trained on the **three-modality dataset**. For pretrained baselines (i.e., PLIP, MLLM4Text-ReID, HAM), we report results tested on the three-modality data without additional training. For pretrained-finetuning combination baselines (i.e., CSDN+RDE), all finetuning strategies are conducted on the **three-modality dataset**. In Tab.1, Tab.2 and Tab.3, we present the **most recent and best-performing pretrained-finetuning combination (RDE+HAM)**. In addition, we supplement comparisons of performance with finetuning applied to all pretrained baselines on SYSU-TVI dataset, demonstrating that our method achieves the best performance under the same training settings:
>
> | Method             | R1 (t2rgb) | mAP (t2rgb) | R1 (t2ir) | mAP (t2ir) |
> | ------------------ | ---------- | ----------- | --------- | ---------- |
> | PLIP+SDM           | 91.88      | 77.85       | 41.97     | 25.08      |
> | MLLM4Text-ReID+RDE | 93.86      | 79.77       | 54.56     | 41.41      |
> | ours | 96.13    | 88.19       | 66.21     | 47.63     |
>
> ***Q4: Color Descriptions in Text Modality (Weakness 3,4,5)***
>
> Please refer to our response to Q3 from reviewer ifeT.
>
> [1] Ye M, Wu Z, Chen C, et al. Channel augmentation for visible-infrared re-identification[J]. IEEE Transactions on Pattern Analysis and Machine Intelligence, 2023, 46(4): 2299-2315.

---

> > ### Author Rebuttal · Reviewer_uFie · 2026-04-03
> >
> > 1:The authors provided a clear mathematical clarification: the orthogonal loss dictates the relationship between the hyperplanes of the two subspaces, while the KL divergence regularizes the internal structure within the subspaces. This explanation is logically sound. Furthermore, the referenced t-SNE visualization confirms empirically that the feature space does not re-entangle. This concern is fully resolved.
> >
> >
> > 2: The authors successfully clarify that their approach does not merely extract single channels from RGB images, but also actively aligns these Channel-Augmented (CA) images with text to simulate variations unrelated to color, thereby helping the model learn color-invariant representations. This is a commendable and effective strategy. However, the rebuttal evades direct acknowledgement of the fundamental physical limitation raised in my initial review: single-channel RGB remains an imperfect physical proxy for infrared (Thermal/NIR). Infrared captures heat radiation or near-infrared reflectance, which behaves entirely differently from visible light reflectance. Nevertheless, I acknowledge the empirical necessity of the authors' design; given that most existing VI-ReID datasets lack image-level paired RGB-IR data, synthesizing paired, color-irrelevant single-channel images from RGB inputs serves as a highly practical and valuable workaround to mitigate this dataset limitation.
> >
> > 3：The authors explicitly clarified that all non-pretrained and fine-tuning combination baselines were indeed trained on the identical three-modality dataset. Furthermore, the supplementary table provided in the rebuttal successfully demonstrates that the proposed method still substantially outperforms other pre-trained combinations (like PLIP+SDM and MLLM4Text-ReID+RDE) under the exact same tri-modal training settings. This completely clears up the experimental ambiguity.

---

> > > ### Author Response · Authors · 2026-04-04
> > >
> > > We sincerely thank Reviewer uFie for the careful and detailed review of our work. We are also pleased that our rebuttal has addressed your concerns. Your comments on both the theoretical and experimental aspects (e.g., theoretical consistency, IR proxy, and baseline fairness) are very helpful in further improving our paper. We will carefully incorporate these discussions to strengthen our final manuscript.
> > >
> > > Thank you again for your valuable time and detailed review!

---

### Decision · Program_Chairs · 2026-04-30

**Decision:**

Accept (regular)

**Comment:**

The reviewers are unanimous in supporting acceptance. In my view, the paper represents for introducing a meaningful new task and benchmark, proposing a well-motivated and technically sound framework for tri-modal semantic alignment, delivering strong empirical results with convincing rebuttal clarifications. Overall, this is a good contribution with both community value and practical relevance. I therefore recommend acceptance, with potential consideration for oral presentation.